# Prediction of Glacially Derived Runoff in the Muzati River Watershed Based on the PSO-LSTM Model



**Xiazi Yang [1], Balati Maihemuti [1,2,\*], Zibibula Simayi [1,2], Muattar Saydi [3] and Lu Na [1]**

[1] School of Geography and Remote Sensing Sciences, Xinjiang University, Urumqi 830046, China; xiazy18@lzu.edu.cn (X.Y.); zibibulla3283@sina.cn (Z.S.); luna100317@163.com (L.N.)

[2] Key Laboratory of Xinjiang General Institutions of Higher Learning for Smart City and Environment Modeling, Xinjiang University, Urumqi 830046, China

[3] School of Geography and Planning, Sun Yat-Sen University, Guangzhou 510275, China; muaitaer@mail.sysu.edu.cn

\* Correspondence: bmaihemuti@xju.edu.cn

**Abstract:** The simulation and prediction of glacially derived runoff are significant for water resource management and sustainable development in water-stressed arid regions. However, the application of a hydrological model in such regions is typically limited by the intricate runoff production mechanism, which is associated with snow and ice melting, and sparse monitoring data over glacierized headwaters. To address these limitations, this study develops a set of mathematical models with a certain physical significance and an efficient particle swarm optimization algorithm by applying long- and short-term memory networks on the glacierized Muzati River basin. First, the trends in the runoff, precipitation, and air temperature are analyzed from 1990 to 2015, and differences in their correlations in this period are exposed. Then, Particle Swarm Optimization–Long Short-Term Memory (PSO-LSTM) and Bi-directional Long Short-Term Memory (BiLSTM) models are combined and applied to the precipitation and air temperature data to predict the glacially derived runoff. The prediction accuracy is validated by the observed runoff at the river outlet at the Pochengzi hydrological station. Finally, two other types of models, the RF (Random Forest) and LSTM (Long Short-Term Memory) models, are constructed to verify the prediction results. The results indicate that the glacially derived runoff is strongly correlated with air temperature and precipitation. However, in the study region over the past 26 years, the air temperature was not obviously increasing, and the precipitation and glacially derived runoff were significantly decreasing. The test results show that the PSO-LSTM and BiLSTM runoff prediction models perform better than the RF and LSTM models in the glacierized Muzati River basin. In the validation period, among all models, the PSO-LSTM model has the smallest mean absolute error and root-mean-square error and the largest coefficient of determination of 6.082, 8.034, and 0.973, respectively. It is followed by the BiLSTM model having a mean absolute error, root-mean-square error, and coefficient of determination of 6.751, 9.083, and 0.972, respectively. These results imply that both the particle swarm optimization algorithm and the bi-directional structure can effectively enhance the prediction accuracy of the baseline LSTM model. The results presented in this study can provide a deeper understanding and a more appropriate method of predicting the glacially derived runoff in glacier-fed river basins.

**Keywords:** glacial hydrology; glacially derived runoff; PSO-LSTM model; BiLSTM model





## 1. Introduction

Glaciers, as one of the main elements of the cryosphere, have been widely studied owing to their high sensitivity to climate changes [1–4] and their significance in the multitude of water usages and ecosystem goods in the glacier-fed downstream areas [5–7]. Approximately 75% of the world's freshwater resources are stored in the cryosphere [8], and glacial meltwater production provides an extremely important water source for the development of oases agroecosystems in drylands globally. The accuracy of its forecasts

is of great significance to the sustainable development of oasis agroecology supported by water resources, such as playing an important role in guiding flood and drought prevention, reservoir scheduling, the optimal allocation of water resources, and the optimization of planting structures. Therefore, it is important to study the impact of glacial meltwater on the sustainable development of oases habitats and to accurately assess changes in glacially derived runoff patterns from aspects of climate change.

A series of methods have been derived in the field of glacier hydrology, such as the direct observation method [9]; the glacier material–energy balance method [10,11]; the water chemistry tracing method [12]; the hydrological modeling method, which is based on a conceptual model with certain physical significance; and the physical model, which is constructed on the basis of the glacier flow production process, such as an energy balance model [13,14], temperature index model, and modified temperature index model [15,16]. However, glacier water production represents a complex process. The precipitation in the glacier area can form surface runoff; snowfall at the right temperature can also form surface runoff. There is also glacier meltwater, which includes the ice meltwater and liquid precipitation on the ice. In the process of glacier melting, the objective existence of rain, snow, and ice transformation also makes the situation more complex. According to theoretical analysis, each hydrological component exists in infiltration, retention, sublimation, evaporation, and other phenomena. Water infiltration forms part of the lateral flow to recharge the aquifer, which may also form the baseflow to recharge the river. The infiltration of water can also form part of the lateral flow to recharge a river, in addition to an aquifer, which may also generate the baseflow to recharge the river, and then deduct the loss along the hydrological station; theoretically, this informs the observed hydrological data. Second, there are uncertainties in physical models, input data, and model parameters, which will eventually lead to poor model accuracy or even cause modeling difficulties.

The process-oriented physical model has good interpretability, but unfortunately, the process-oriented physical model needs a large amount of basic data, such as soil water storage capacity and hydraulic conductivity. At the same time, high-coverage glaciers are often located at high latitudes and high altitudes, where climatic conditions are harsh and the areas sparsely populated; the distribution of observation sites is limited and extremely heterogeneous; and the observation elements are limited. Thus, the application of process-oriented physical models is greatly limited in small watershed areas where there is a lack of actual measurement data. Given that few previous studies have applied artificial neural network models to special high-coverage glacial basins—although a number of studies have used artificial neural network models for hydrologic forecasting—most of them are based on precipitation-driven runoff studies, i.e., precipitation as input and runoff as output; or the models are determined simply from recorded single hydrologic time series data, following the correlation and biased autocorrelation of previous hydrologic series data input steps, nodes, and other variables, and no other input feature data. In this paper, we use air temperature and precipitation as input feature data to test whether LSTM can predict the glacial runoff process in the medium and long term. The prediction accuracy under hyperparametric optimization (PSO-LSTM) and structural optimization (BiLSTM) is also compared with that of the benchmark LSTM model, and the aim is to investigate the proposed improved model with higher accuracy against the benchmark model.

An artificial neural network can be understood as a data-driven black-box model of a nonlinear relationship between input and output data represented through a feedforward neural network structure. Using artificial neural networks eliminates the need for knowledge about the relevant physical parameters of a watershed, and they have been widely used in hydrological research in recent years [17]. For instance, Kratzert [18] used the LSTM model for 241 watersheds in the CAMELS dataset in the United States; Kratzert showed that the LSTM could simulate snowmelt runoff better than the RNN in snowfall-affected watersheds and that simulation results were comparable to those of the SAC-SMA-Snow-17 model with a physical basis. LSTM models are strongly influenced by hyperparameters, and the random initialization parameters and the determination and combination of impor-

tant hyperparameters lead to poor results of the models sometimes; therefore, this paper proposes the PSO-LSTM method, and the combination of the two methods can retain the excellent training as well as prediction performance of LSTM, but can also optimize its important hyperparameters adaptively. For example, Gharabaghi et al. proposed a new optimization algorithm PSOGA by combining the particle swarm optimization algorithm PSO with the genetic algorithm GA and applied the optimization algorithm to the adaptive fuzzy system ANFIS model, and the results of that study showed that the accuracy of the ANFIS model mixed with PSOGA is better than that of the individual algorithm [19]. Mohammadi et al. proposed to combine the multilayer perceptron model mixed with the particle swarm algorithm PSO and then integrated with the differential evolution algorithm DE to obtain the MLP-PSODE hybrid model, which was found to have a more efficient structure and less artificial settings for hyperparameters in predicting SSL tasks [20]. In other studies, such as one combining PSO with SVR for ammonia nitrogen prediction [21] or another combining PSO with an ANN model for early detection of dengue fever disease [22], the PSO optimization algorithm has the effect of improving the accuracy of the model (JiAN ZHOU). Secondly, this paper also proposes the BiLSTM model: although the LSTM model is more suitable than the traditional RNN for dealing with long time series data, both networks can only process data in one direction, i.e., they can only rely on the data of the previous moment to predict the next one, which tends to ignore the information of the future moment. To solve the above problems, this paper also proposes to model the BiLSTM network, which is composed of two LSTM neural networks, forward and backward, which not only can obtain past information of the input data but can also use future information, which is helpful for solving the task of sequential data. For example, it has high-accuracy applications in urban flood process forecasting [23], monthly average groundwater level fluctuation prediction [24], and time-by-time soil temperature prediction [25].

In the above literature, the hybrid model can well improve the prediction performance of the data in the experiments, so it is necessary to select a suitable optimization method for determining the optimal parameters of the model and optimizing the model structure when determining the hybrid model. In recent years, neural network models have been successfully applied to the fields of flood control, sedimentation, and water quality, achieving good results [26–28]. However, there have been fewer studies related to the application of artificial neural network models to alpine glacier basins with complex hydrological mechanisms, such as Ji in the alpine glacier basin of the Kumaric and Toxkan rivers [29]. The application of the LSTM models in alpine glacier areas has been a major step forward in hydrological research. A study on monthly runoff simulation in a Tibetan Plateau permafrost basin showed that a two-dimensional input (air temperature and precipitation) has been promising for simulating and predicting runoff changes during permafrost basin runoff [30]. In the LSTM neural network models, air temperature and rainfall are the data drivers, and these models are ideal to study alpine glacier areas, especially in scarce data regions. Since runoff from alpine glaciers is dominated by meteorological factors, using meteorological data as input data of a neural network model has proven to be sufficient to achieve an excellent prediction performance.

The Muzati River's runoff, monitored at the Pochengzi hydrological station, is the largest of the five tributaries of the Weigan River, which originates from the Khan Tengri peak at the southern foot of the Tianshan Mountains in Central Asia, flows downward through the Baicheng oases basin (i.e., Baicheng County), and finally disappears into the Wei-Ku oasis (i.e., Kuqa County, Xinhe County, and Shaya County). The Baicheng oases ecological units have high mountainous conditions with abundant precipitation and low temperatures and have developed a large number of modern glaciers above the headstreams of the Muzati River basin, with a glacier coverage of approximately 48.2% and snow and ice meltwater proportion of approximately 93% [31]. The ecological carrying capacity of water resources in the Wei-Ku and Baicheng oases is quite limited, especially during dry periods with sparse precipitation and extreme evapotranspiration (ET), and

glacially derived runoff has an essential effect on the water resource management, which influences local ecological sustainability [32]. Therefore, the analysis and prediction of glacially derived runoff in this study area are representative for studying the response of glacially derived runoff in the context of climate influence and regulating the sustainable development of an oasis under water resource constraints. The Pochengzi hydrological station is located at the river outlet and has long-term observation records and quality data to initiate a hydrological model.

The main contributions of this study are as follows:

(1) To determine whether the LSTM can predict the glacier runoff process in the medium and long term, and to compare the prediction accuracy with the benchmark LSTM model under hyperparametric optimization (PSO-LSTM) and structural optimization (BiLSTM), with the aim of proposing a hybrid improvement model with higher accuracy based on the original model.

(2) To carry out medium- and long-term hydrological forecasting to provide important guidance for reservoir scheduling, the optimal allocation of water resources, and the optimization of planting structures and for the sustainable development of oasis agroecology supported by water resources.

## 2. Study Area Overview and Data Sources

### 2.1. Study Area

The headwater glacier area ($80°20'0''$–$81°0'0''$E, $41°50'0''$–$42°20'0''$N) controlled by the Pochengzi hydrological station is shown in Figure 1. This glacier area is located at the southern foot of the Tianshan glaciers in Central Asia, with a higher terrain in the northwest and a lower terrain in the northeast, having an average elevation of 3912 m. Generally, water vapor fluxes over this glacier area mainly originate from the westerly jet stream and the Arctic Ocean's cold flow [33,34]. The area covered by snow and ice accounts for approximately 48.2% of the total study area, and the contribution of glacial meltwater to the runoff at the river outlet is 93% [31]. The natural climate and geographical conditions are slightly affected by local anthropogenic influence but are highly influenced by global climate changes. The average multi-year runoff monitored at the Pochengzi hydrological station is $14.5 \times 10^8$ m$^3$, and it consists mainly of snow and ice meltwater. The runoff is mainly represented in summer (i.e., from June to August), when the temperature is the highest. The glaciers at the headwaters of the Pochengzi hydrological station are extremely sensitive to climate changes. Therefore, it is important to realize an accurate glacier hydrological simulation and prediction in this glacier-fed river basin. In this way, the intensity of glacier response to climate changes can be clarified, and the subsequent findings can favor the sustainable development of water resources in water-sparse oases.

### 2.2. Data Sources and Processing

The digital elevation model (DEM) data were obtained from the Geospatial Data Cloud of the Chinese Academy of Sciences, while meteorological data (monthly average air temperature and precipitation) were acquired using the GEE platform for ERA5-Land reanalysis data. The ERA5-Land has been produced by replaying the land component of the ECMWF ERA5 climate reanalysis. Reanalysis combines model data with observation data from across the world into a globally complete and consistent dataset using the laws of physics [35]. The spatial resolution of the collected data was about 11 km × 11 km. The study area contained approximately 22 effective meteorological grid points. The glacier data were obtained from the Second Chinese Glacier Inventory using an area mask extraction method.

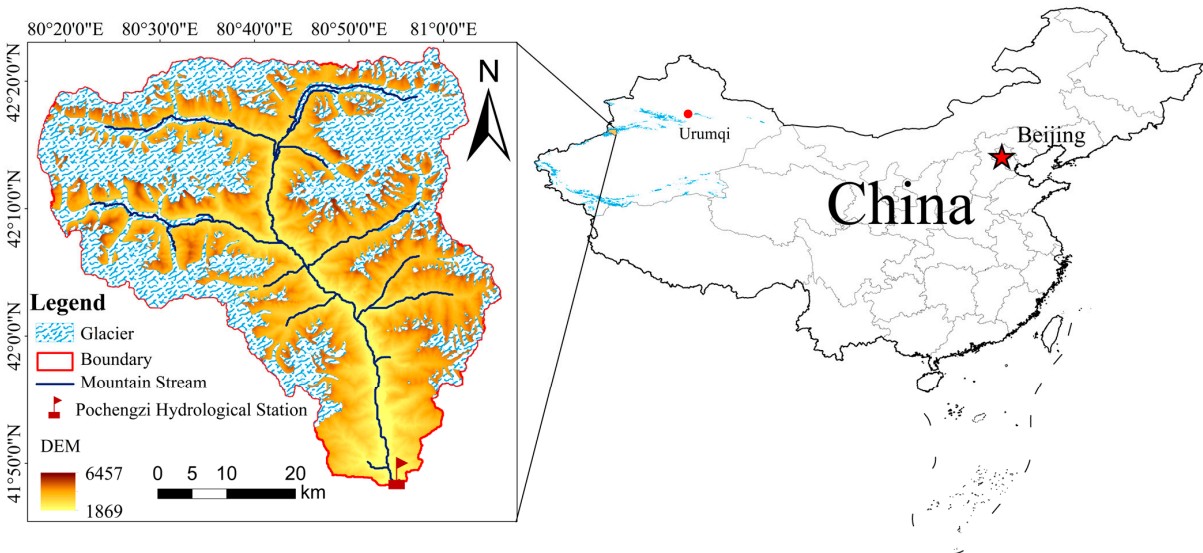

**Figure 1.** Location and glacier distribution in Muzati river basin.

## 3. Materials and Methods

### 3.1. Statistical Analysis

Numerous studies have demonstrated that glacier ablation is strongly influenced by climate and particularly sensitive to air temperature changes [1,2]. In this study, two climate factors, the air temperature and the precipitation, were considered due to their physical significance in the fields of glacier hydrology and climate change [16,36]. The trends in runoff, precipitation, and air temperature, as well as their significance, over the last 26 years were analyzed using time series observations. Two statistical methods, the Spearman rank correlation test and the Mann–Kendall trend test, were comparatively performed to test the trends in runoff, precipitation, and air temperature and their significance in each time series observation. More details of these two methods can be found in [37,38].

### 3.2. LSTM Hydrological Prediction Model Optimized by PSO Algorithm

#### 3.2.1. PSO Algorithm

The idea of the PSO algorithm was developed based on the process of group migration and foraging activities in the animal kingdom [39], and its core idea is that in group activities, each individual is represented. In the PSO algorithm, an individual is equivalent to a particle, the location of the food represents an optimal solution to the considered problem, and the distance between a particle and the optimal solution denotes the value of the current particle's objective function. Individual particles benefit from the experience discovered and accumulated by the particle swarm in this process while finding an optimal solution to the problem through swarm collaboration [40]. The PSO algorithm, which evolved from the mentioned idea, is a stochastic search algorithm that simulates biological activities in nature and is also a branch of evolutionary computation.

The principle of the PSO algorithm is as follows. The PSO is initialized as a group of random particles, each of which has two attributes: position $x$ and velocity $v$. Subsequently, an optimal solution is found through iterations, and in each iteration, the particle follows two extremes to update its attributes; the first extremum is the optimal solution *pbest* found by the particle itself, and the other extremum is the optimal solution found by the entire particle population, i.e., the global optimal solution *gbest* [41]. The velocity and position of a particle are updated as follows:

$$v_{i+1} = w \times v_i + c_1 \times rand_1 \times (pbest_i - x_i) + c_2 \times rand_2 \times (gbest_i - x_i), \quad (1)$$

$$x_{i+1} = x_i + v_{i+1}, \quad (2)$$

where $w$ is the inertia factor, which controls the weight distribution of particles in *pbest* and *gbest*; $c_1$ and $c_2$ are learning factors, which are used to adjust the flight step; $rand_1$ and $rand_2$ are random numbers between zero and one; $v_i$ and $x_i$ represent the velocity and position of the *i*th dimension of a particle, respectively; $pbest_i$ and $gbest_i$ are the local and global optimal solutions of the *i*th dimension of the optimal particle position, respectively.

### 3.2.2. LSTM Networks

The LSTM is a special type of a recurrent neural network (RNN), and compared with the traditional RNN, the basic concept of the LSTM is the addition of the cell state and gate structure, as shown in Figure 2. The gate structure enables the addition and removal of information so that the structure learns the information that should be saved or forgotten during the training process, which represents a hyperparameter-controlled LSTM. The hyperparameter-controlled LSTM neural network structure can achieve good performance on long sequences. However, the setting of hyperparameters brings a large uncertainty to the model's simulation accuracy [26], so the PSO particle swarm optimization algorithm is combined with LSTM networks to obtain the optimal hyperparameters of the model.

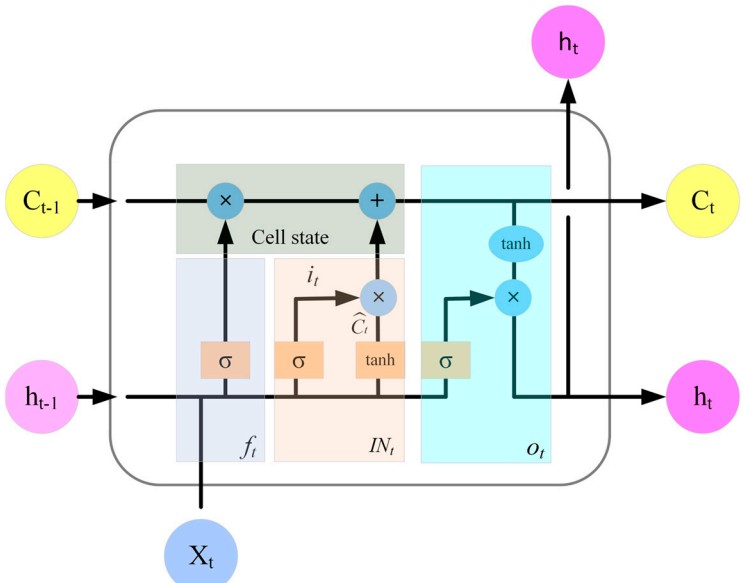

**Figure 2.** The basic structure of Long-Short Term Memory (LSTM) at time step t. $f_t$, forget gate; $IN_t$, input gate; $O_t$, output gate, and cell state. The $C_t$ value represents the cell state at time step t; $C_{t-1}$ value represents the cell state at time step t − 1; $X_t$, current inputs; $h_t$, output result of hidden layer at time step t; $h_{t-1}$, output result of hidden layer at time step t − 1; $\sigma$, sigmoid activation function; tanh, hyperbolic tangent function.

The input and output data of the LSTM model at moment *t* are shown in Figure 2, where it can be seen that the LSTM network consists of the forgetting gate $f_t$, input gate $IN_t$, output gate $O_t$, and cell state, which is adjusted through the internal gate structure. The input gate $IN_t$, also known as the update gate layer, includes $i_t$ and $\hat{C}_t$, which work together to determine the information that needs to be updated in the cell state at time $t$. At the same time, the output gate $O_t$ updates the cell state and decides the output. The input data of the LSTM network at moment *t* includes three parameters: the cell state $C_{t-1}$, the hidden layer state $h_{t-1}$, and the input vector $x_t$ at moment t; the output consists of two parameters, the cell state $C_t$ and the hidden layer state $h_t$; $h_t$ is also used as the output at moment *t*. The air temperature and precipitation data of the input model were normalized and fed to the model; the output data of the model were denormalized to obtain the output of the simulated flow. The LSTM algorithm was implemented as follows:

(1) The gate layer $f_t$ removes information on the $(t − 1)$ moment cell state:

$$f_t = \sigma\left[W_f \times (h_{t-1}, x_t) + b_f\right].\tag{3}$$

(2) The input gate $IN_t$, also known as the update gate layer, consists of two parts, $i_t$ and $\hat{C}_t$, which are respectively given by:

$$i_t = \sigma[W_i \times (h_{t-1}, x_t) + b_i],\tag{4}$$

$$\hat{C}_t = tanh[W_c \times (h_{t-1}, x_t) + b_c],\tag{5}$$

The cell state $C_t$ is:

$$C_t = f_t \times C_{t-1} + i_t \times \hat{C}_t.\tag{6}$$

(3) The output gate layer $O_t$ calculates the output values of the cell state $C_t$ and the hidden layer state $h_t$ as follows:

$$O_t = \sigma[W_o \times (h_{t-1}, x_t) + b_o],\tag{7}$$

$$h_t = O_t \times tanh(C_t).\tag{8}$$

### 3.2.3. LSTM Model Optimized by PSO

As mentioned above, the LSTM model adopts the cell state and gate structure and avoids the problems of traditional RNNs, which are strongly affected by the short-term memory and disappearance of gradients during backpropagation. However, the introduction of the gate structure increases the number of parameters, hyperparameters, and gates. Moreover, the parameter settings define the effect of the model. Aiming to adjust the LSTM model parameters to optimize its prediction effect, the PSO algorithm is used to optimize the number of neurons in the hidden layer of the LSTM model, learning rate, and other model parameters to obtain the optimal combination of model parameters. The PSO-LSTM model construction process is shown in Figure 3.

### 3.3. Model Validation

The coefficient of determination $R^2$, the Nash–Sutcliffe efficiency coefficient NSE, and the root-mean-square error RMSE were used in this study to evaluate the simulation effect. The model effect is optimal when the absolute values of $R^2$ and NSE are between zero and one and the RMSE is closer to zero. The evaluation indexes are calculated as follows:

$$\text{RMSE} = \sqrt{\frac{1}{n}\sum_{i=1}^{n}(Q_{obs,i} - Q_{sim,i})^2}\tag{9}$$

$$\text{MAE} = \frac{1}{n}\sum_{i=1}^{n}\left|Q_{obs,i} - Q_{sim,i}\right|\tag{10}$$

$$R^2 = \left[\frac{\sum_{i=1}^{n}(Q_{obs,i} - Q_{obs,a})(Q_{sim,i} - Q_{sim,a})}{\sqrt{\sum_{i=1}^{n}(Q_{obs,i} - Q_{obs,a})^2}\sqrt{\sum_{i=1}^{n}(Q_{sim,i} - Q_{sim,a})^2}}\right]^2\tag{11}$$

where $Q_{obs,i}$ is the measured flow value at moment $i$; $Q_{sim,i}$ is the simulated flow value at moment $i$; $Q_{obs,a}$ is the average value of measured flow; $Q_{sim,a}$ is the average value of simulated flow.

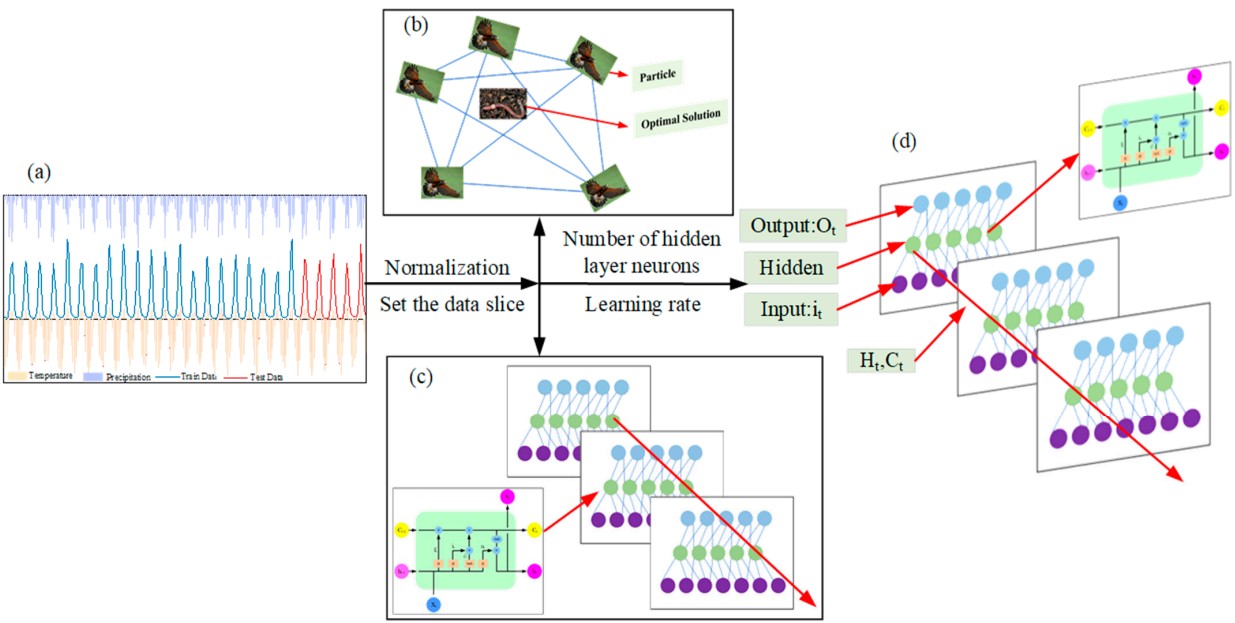

**Figure 3.** PSO-LSTM model construction process. (**a**) Data preparation, mainly including data standardization and slicing; (**b**) Schematic diagram of the social network structure of the PSO algorithm; (**c**) Schematic diagram of the initial LSTM model structure; (**d**) The PSO-LSTM model after parameter optimization.

## 4. Results

### 4.1. Effects of Glacier Discharge and Climatic Factors

The average monthly value of the monitored flow at the hydrological control station of Pochengzi and the average monthly air temperature and precipitation in its upstream glacier area are presented in Figure 4. As shown in Figure 4, the flow, air temperature, and precipitation have strong intra-annual characteristics, and they all show single-peak characteristics during the year, having larger values from May to September, especially in July and August. The air temperature of the glacier area upstream of the Pochengzi hydrological station is mostly below zero Celsius degrees and only slightly above zero Celsius degrees in the peak months of the year. Figure 4 shows that the peak of the flow coincides well with the peaks of air temperature and precipitation.

The Spearman rank correlation test and M-K rank correlation test were performed to test the trends and significance of the air temperature and precipitation in the glacier area. The glacially derived runoff monitored at the Pochengzi hydrological station over the past 26 years was analyzed, and the results are shown in Table 1.

**Table 1.** Average monthly trends of the runoff, air temperature, and precipitation from 1990 to 2015 at Pochengzi hydrological station and its upstream watershed.

| Parameter | Spearman Rank Correlation Test | | | | | M-K Rank Correlation Test | | | |
|---|---|---|---|---|---|---|---|---|---|
| | $r_s$ | \|T\| | $T_{\alpha/2}$ | Trend | Significance | Z | $Z_{\alpha/2}$ | Trend | Significance |
| Runoff | −0.013 | 0.220 | 1.651 | decreasing | slight | −0.358 | 1.960 | decreasing | slight |
| Air Temperature | 0.023 | 0.395 | 1.651 | increasing | slight | 0.415 | 1.960 | increasing | slight |
| Precipitation | −0.013 | 0.233 | 1.651 | decreasing | slight | −0.198 | 1.960 | decreasing | slight |

Note: $r_s$ is the Spearman rank correlation coefficient; |T| is the *t*-test statistic; $T_{\alpha/2}$ is the critical value at the 95% confidence level in the *t*-test; Z is the M-K rank correlation test statistic; $Z_{\alpha/2}$ is the critical value at the 95% confidence level.

As shown in Table 1, the monthly trends of the runoff and upstream watershed air temperature and precipitation monitored at the Pochengzi hydrological station over the past

26 years indicate that the glacier runoff and watershed precipitation had an insignificant decreasing trend, while the air temperature showed an insignificant increasing trend. To understand the glacier runoff driven by the air temperature and precipitation in the alpine glacier area, it is necessary to establish a set of mathematical models with certain physical meaning, which should be simple and feasible. In view of this, this study proposed a PSO-LSTM hydrological model that uses the air temperature and precipitation as input data and the glacial runoff as output data. Using such input and output data increases the physical significance of the model. In addition, the proposed model indicates the relationships between the input and output data, which represents the mapping of the response of complex glacier hydrological processes to natural climate changes, and finally adjusts the parameters to obtain the global optimal solution to realize the glacier hydrology. The proposed model was adjusted to obtain the global optimal solution to realize the simulation of glacier hydrology.

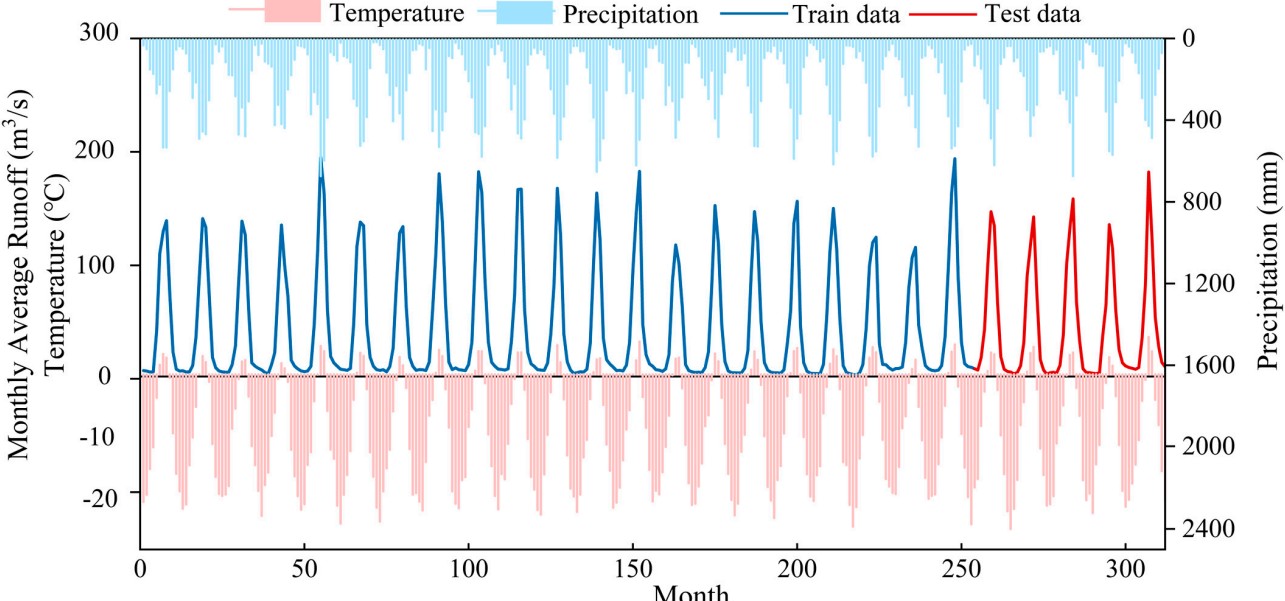

**Figure 4.** Average monthly runoff, air temperature, and precipitation at Pochengzi hydrological station from 1990–2015.

### 4.2. Model Performance Evaluation

An RF regression model consists of multiple decision trees with no or very weak mutual correlation. The RF method randomly selects samples from the training set as the root node samples of the trees and uses a certain number of feature attributes when building each regression tree. This method performs node splitting [42], constructs a single regression decision tree, and repeats these steps to obtain multiple regression decision trees; the final prediction results denote averaged regression decision tree prediction results.

The LSTM network constructed in this study consisted of a single LSTM layer and a fully connected layer; the batch size was set to eight; Adam optimizer was used as a network optimizer. The number of memory units in the LSTM hidden layer was 50; the learning rate was set to 0.01 [43]; the maximum number of iterations in the model training was set to 500 to avoid the phenomenon of overfitting when the amount of data was small. The LSTM model adopted the dropout strategy in the training process, and the dropout value was set to 0.3 in the modeling process. The dropout strategy was used to weaken the association between network nodes during the training process and to reduce the dependence of the network on individual neurons, thus enhancing the generalization ability to overcome the overfitting phenomenon. Because the initial values of model parameters have a certain influence on the model's performance, causing slight differences in the prediction results, the LSTM network prediction model was tested 20 times using different model parameters.

The mean values of the RMSE, MAE, and $R^2$ evaluation indicators of the 20 experimental tests were calculated, and a group of parameters with similar mean values of the evaluation metrics were selected to construct a benchmark LSTM model.

In the construction process of the PSO-LSTM hybrid model, the learning rate and the number of nodes in the hidden layer of the benchmark model were optimized using the PSO algorithm. The learning rate was in the range of 0.001–0.1, and the number of nodes in the hidden layer was in the range of 10–200. The parameters of the PSO algorithm were set as follows: the number of particle populations was 5; the acceleration factors were $C_1 = C_2 = 2$; the maximum and minimum inertia weight values were 1.2 and 0.8, respectively; and the rest of the parameters were set as default. The optimal model parameters were obtained by the PSO algorithm and substituted into the LSTM model for model training; the model simulation results were recorded.

Since the operation of the LSTM model is unidirectional, this study introduced the BiLSTM model. The BiLSTM model is a combination of the forward and backward LSTM models and contains richer information because its output includes the information of both forward and backward LSTMs. For consistency, in this study, the LSTM model settings in the BiLSTM model were kept consistent with that of the benchmark LSTM model.

The structural parameters of the benchmark LSTM model, PSO-LSTM model, and BiLSTM model are shown in Table 2, where $L_i$ denotes the number of LSTM layers, $D_i$ represents the number of fully connected layers, and $B_i$ is the number of BiLSTM layers; $L_n$ denotes the number of LSTM hidden-layer nodes, $D_n$ is the number of dense cells, and $B_n$ represents the number of BiLSTM model layer cells.

**Table 2.** Structure of the monthly average runoff prediction model.

| Model | Hidden Layer | Hidden Layer Setting | Dropout Value | Optimization Function | Batch Size |
|---|---|---|---|---|---|
| Benchmark LSTM | $L_i = 1, D_i = 1$ | $L_n = 50, D_n = 1$ | 0.3 | Adam optimizer | 8 |
| PSO-LSTM | $L_i = 1, D_i = 1$ | $L_n = 50, D_n = 1$ | 0.3 | Adam optimizer | 8 |
| BiLSTM | $B_i = 1, D_i = 1$ | $B_n = 50, D_n = 1$ | 0.3 | Adam optimizer | 8 |

The results of the four models are shown in Figures 5 and 6 and Table 3. Figure 5 shows the fitted curves of the predicted and measured values of the RF, LSTM, BiLSTM, and PSO-LSTM models in the training period. In the model training period, the lowest $R^2$ was 0.974 (the RF model), and the highest $R^2$ was 0.994 (the PSO-LSTM model), which was followed by the $R^2$ value of 0.990 achieved by the LSTM model and the $R^2$ value of 0.976 obtained by the BiLSTM model. Figure 6 shows the fitted curves of the predicted and measured values of the four models in the validation period. The lowest $R^2$ of 0.953 was achieved by the RF model, while the highest $R^2$ was 0.973, and it was obtained by the PSO-LSTM. The $R^2$ value of the BiLSTM model was 0.972, and the $R^2$ value of the LSTM model was 0.955.

Relative to the benchmark LSTM model, the prediction accuracy of both the PSO-LSTM model and the BiLSTM improved to different degrees, and the root-mean-square error RMSE of the PSO-LSTM model and BiLSTM model in the validation period was reduced by 23.79% and 13.84%, respectively; the mean absolute error MAE was reduced by 13.74% and 4.25%, respectively.

According to the statistics of the evaluation parameters of the four models in Table 3, in the validation period, the PSO-LSTM and BiLSTM models had better performance than the other two models in terms of each evaluation index, which proved that the PSO algorithm had a positive effect on parameter tuning. This also demonstrated that the bidirectional LSTM structure was effective in reducing model errors and improving simulation accuracy. Figure 7 shows the fitted curves of the predicted and measured values of the four models in the validation period.

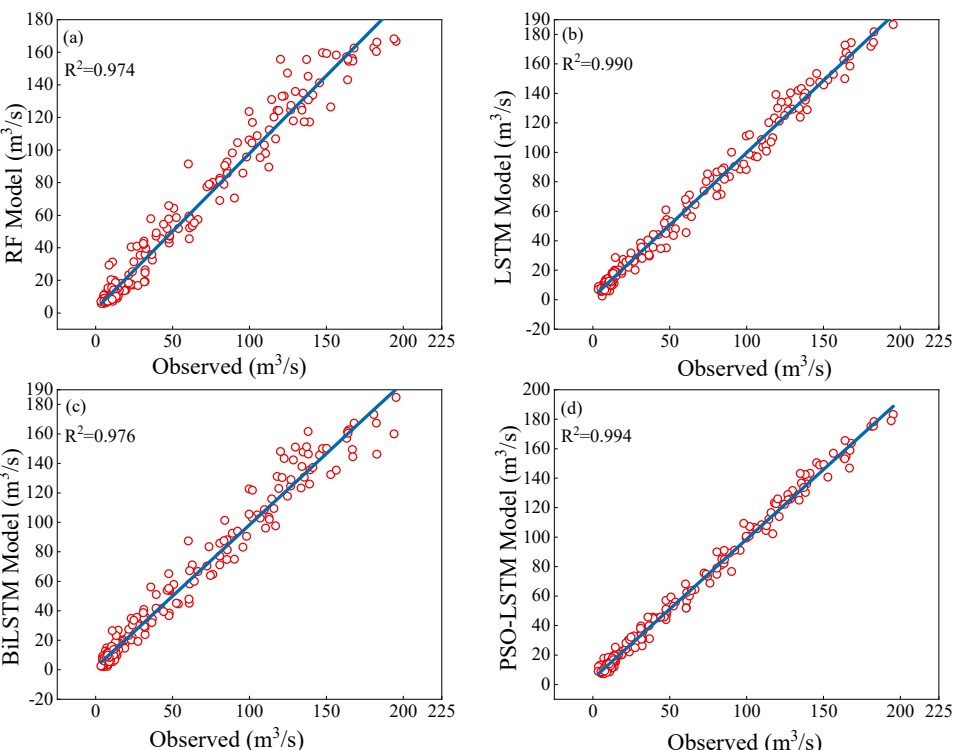

**Figure 5.** Scatter plots and fitting lines of simulated and observed monthly average runoff for the four models during the training period, with $R^2$ representing the coefficient of determination. (**a**) RF model; (**b**) LSTM model; (**c**) BiLSTM model; (**d**) PSO-LSTM model.

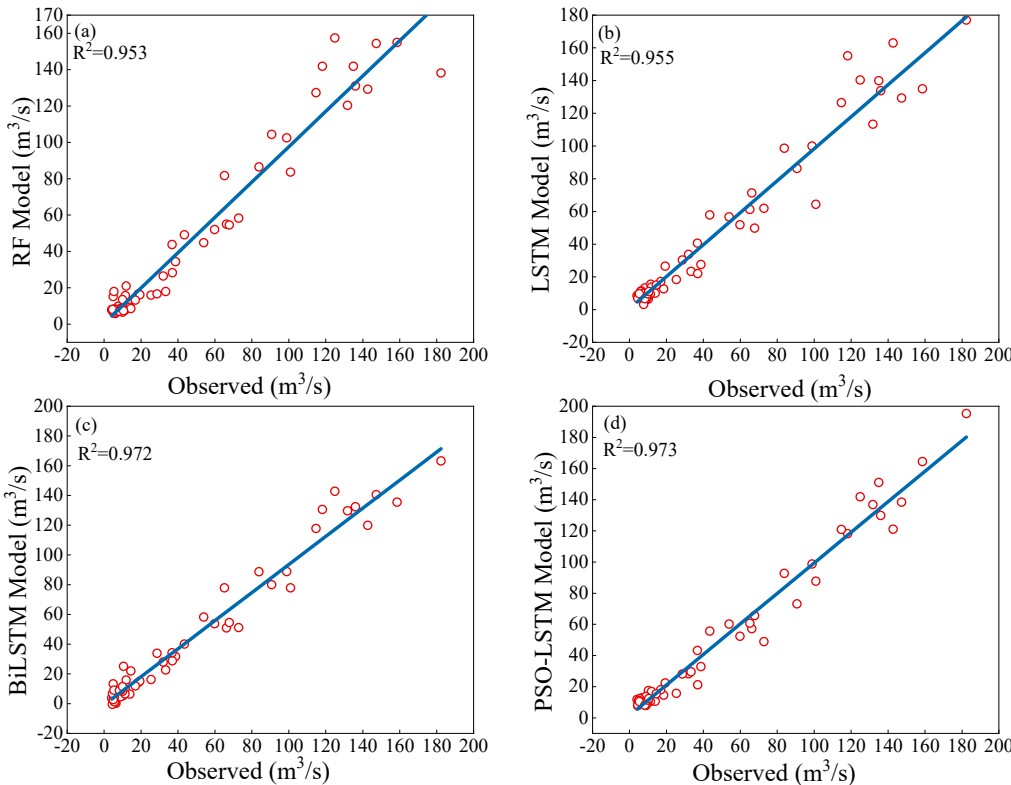

**Figure 6.** Scatter plots and fitting lines of simulated and observed monthly average runoff for the four models during the test period, with $R^2$ representing the coefficient of determination. (**a**) RF model; (**b**) LSTM model; (**c**) BiLSTM model; (**d**) PSO-LSTM model.

**Table 3.** Evaluation results of the RF, LSTM, BiLSTM and PSO-LSTM models in monthly average runoff simulation during the training period (1990–2010) and testing period (2011–2015) in the mu-zati river basins.

| Period | Model | RMSE | $R^2$ | MAE |
|--------|-------|------|-------|-----|
| Training | RF | 8.459 | 0.974 | 5.433 |
| | LSTM | 5.048 | 0.990 | 3.746 |
| | BiLSTM | 8.083 | 0.976 | 5.359 |
| | PSO-LSTM | 4.889 | 0.994 | 3.867 |
| Validation | RF | 10.756 | 0.953 | 7.430 |
| | LSTM | 10.542 | 0.955 | 7.051 |
| | BiLSTM | 9.083 | 0.972 | 6.751 |
| | PSO-LSTM | 8.034 | 0.973 | 6.082 |

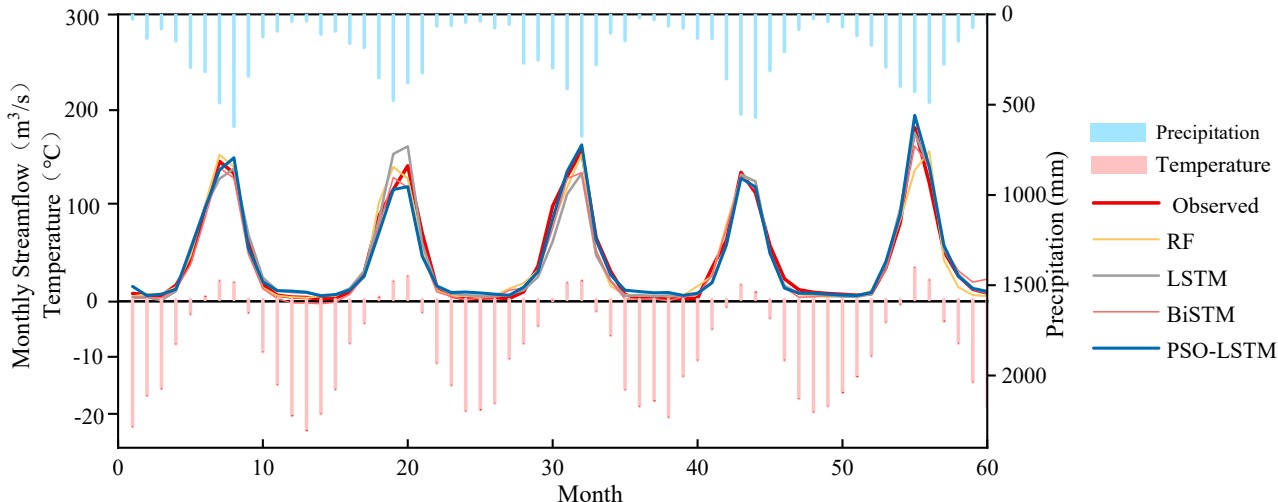

**Figure 7.** Simulation results of monthly average runoff processes during the test period.

### 4.3. LOSS and RMSE Results in Training Period

The loss function curve and the changing trend of the RMSE value of the LSTM model in the training period are presented in Figure 8. In the first 10 training epochs, the loss rapidly decreased, and after 100 training epochs, the loss gradually decreased; after 500 epochs, both the loss function and RMSE tended to become stable. Since the model was trained using the normalized training data, the loss function and RMSE in Figure 8 are presented in the normalized form.

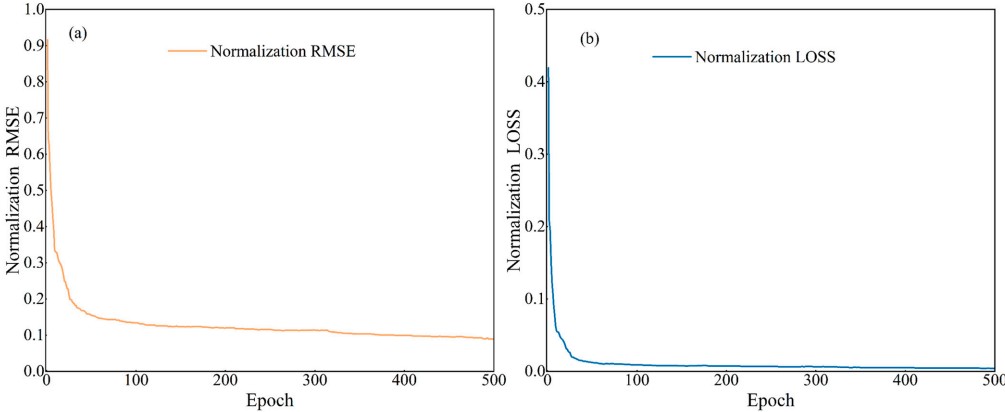

**Figure 8.** The training period results of LSTM model at Pochengzi hydrological station. (**a**) normalized RMSE and (**b**) normalized loss function.

## 5. Discussion

### 5.1. Responses of Glacier Runoff to Climate Changes

Since alpine glaciers are very sensitive to climate factors, especially temperature changes [44], this study quantitatively analyzed the relationships between air temperature, precipitation, and runoff in the past 26 years in the Muzati River basin. The analysis results showed that both precipitation and air temperature are positively correlated with glacially derived runoff, having Pearson correlations of 0.885 and 0.872, respectively. Previous studies have highlighted a strong positive relationship between the air temperature and the glacier runoff and have confirmed that glaciers are natural indicators of air temperature changes [45,46]. Therefore, alterations in glacier and glacier-fed streams should be quantified in a timely manner, as they have strong implications for water resources in arid regions under warming [47,48]. The analysis of the average monthly values of the air temperature, precipitation, and runoff showed that the air temperature has an insignificant upward trend, precipitation shows an insignificant downward trend, and glacially recharged runoff shows a slight overall decrease. The same phenomenon and conclusion have been demonstrated in studies on glacial runoff of different scales in various regions. For instance, Adilai [49] integrated an unmanned aerial vehicle (UAV) and remote-sensing technology to analyze the monthly and interannual trends of the river flow of 10 river sections in the East Pamir Plateau from 1999 to 2020 using a water balance model. The results indicated that the air temperature of the basin in all 10 sections increased with an average increase of approximately 0.016 °C/year; the precipitation increased in the East Pamir region because of the influence of the westerly wind belt. However, the increase in the air temperature and rainfall did not increase the runoff; instead, the flow in all 10 river sections showed varying degrees of decline, with an average value of −21.05%. ANNE's study on a small glacier basin on the flanks of Mount Hood in Oregon, USA, has demonstrated that, although warmer temperatures lead to rapid glacier melting and thus an increased runoff, the consequences of simultaneous glacier retreat overcome this effect, ultimately leading to a significant reduction in the runoff; the impact of the glacier retreat on the runoff at the basin scale is a problem that affects water management [9]. This explains the complex nonlinear relationship between the glacially derived runoff and climate changes, which involves the accumulation, transformation, and ablation of materials, energy transformation, and different contributions of rivers through precipitation and glacier-snow meltwater [50,51]. In the Tien Shan glacial zone, where the study basin of this work is located, Adilai [52] used drone monitoring and satellite remote sensing to study a total of 19 river sections in the middle and west sections of the Tianshan Mountains and found that the monthly runoff of snow- and ice-dominated river sections decreased by an average of 2.46%, and the decline (approximately −4.98 $km^2$/year) or even disappearance of mountain glaciers was the main reason for the decrease in runoff.

In the considered study area, there is a slight growing trend of the air temperature; owing to the special characteristics of the alpine glacier area, the average multi-year temperature is −8.82 °C; the average temperatures in spring, autumn, and winter are 13 °C, −9.29 °C, and −18.81 °C, respectively; the average air temperature in the period from June to August is 1.96 °C. Except for summer, in the other seasons, even when the air temperature increases, the elevated air temperature is below zero, and the effect on glacier melting is negligible. The M-K rank correlation test results of summer air temperatures in the study area revealed no statistically significant changes. Therefore, combined with the slight decrease in the rainfall, the overall performance of the runoff at the Pochengzi hydrological station is relatively stable, showing a slightly decreasing trend. Owing to the characteristics of flow production in alpine glacier areas, it is necessary to pay continuous attention to the glacier runoff response to climate changes.

### 5.2. Prospect of Hydrologic Application of PSO-LSTM and BiLSTM Models

Since temperature and precipitation are considered to be the most critical factors influencing the accumulation and melting of glaciers and snowpack—although only the

long-term monthly average flow rate can be used for prediction—if the random, nonlinear temperature or rainfall changes suddenly or with a trend in the medium- and long-term hydrological prediction, it will cause changes in runoff. If only the long-term monthly average flow rate is used for prediction, it will not be able to capture the trend of runoff changes efficiently, resulting in large deviations in the medium- and long-term forecasts, which will adversely affect the downstream reservoir scheduling, water allocation, and the optimization of the planting structure in the medium and long term. The neural network model can efficiently capture the nonlinear relationship between temperature, precipitation, and runoff, improve the accuracy of runoff prediction based on historical data and mapping the relationship between input and output data, and provide reliable forecast information for medium- and long-term water resource management.

Many recent studies on hydrology and water resources have applied neural network models and machine learning-based models, thus further expanding their application to related fields of research, such as water quality assessment and flood forecasting. There have been a number of studies comparing the simulation effects of the physical models and neural network models, and significant results have been achieved [43,53,54]. Kratzert [18] studied the LSTM with the SAC-SMA-Snow-17 model based on physical processes in a similar research area as ours and found that the simulation effect of the neural network model was not inferior to that of the physical model. Ji [29] studied the LSTM and modified the SWAT model by adding the glacier module to the permafrost region; the results indicated that the prediction accuracy of the neural network model was better than that of the other models. Yuanhao Xu [55] constructed a precipitation-driven PSO-LSTM runoff model to simulate an improved physical process-based SAC-SMA model.

However, the model hyperparameters have an important effect on a model's prediction accuracy. Therefore, this study not only introduced the LSTM neural network model to the typical alpine glacier area but also used the PSO algorithm (PSO-LSTM) to optimize the important hyperparameters of the LSTM model and the BiLSTM neural network structure to optimize the LSTM model structure.

The proposed neural network model does not require much measured hydrological and meteorological data, models with dynamic adaptive features, or an understanding of the complex physical mechanisms of glacier runoff formation, such as the interaction between surface water and groundwater in the permafrost zone and aquifers in the permafrost zone. In addition, to overcome the limitations of numerical runoff simulations under the condition of scarce data, this work proposed using a neural network model in the alpine glacier zone, and the obtained results demonstrate that the proposed neural network model has high prediction accuracy in the monthly runoff simulations in the glacier permafrost zone [56]. In this study, four models, the RF, LSTM, BiLSTM, and PSO-LSTM models, were constructed and compared. The comparison results showed that all models have excellent prediction effects. Moreover, in this study, two important parameters, the number of hidden-layer nodes and the learning rate of the LSTM model, were optimized by the PSO algorithm [57], and the optimized BiLSTM model was compared with the benchmark LSTM model. The results showed that the PSO algorithm-based model parameter optimization and the BiLSTM structure can improve the prediction performance of the benchmark LSTM model. Both the parameter-optimized PSO-LSTM and the structure-optimized BiLSTM have a smaller mean absolute error and root-mean-square error and a larger correlation coefficient between the measured and simulated values than the benchmark LSTM model. The comprehensive evaluation results of each of the four models are shown in the Taylor diagram in Figure 9, where red dots indicate different models, horizontal and vertical axes denote the standard deviation, the radial line represents the correlation coefficient, and the dashed green line shows the root-mean-square error. The advantage of this diagram is that the relationship between the evaluation indexes of a model can be compared from multiple aspects [58,59].

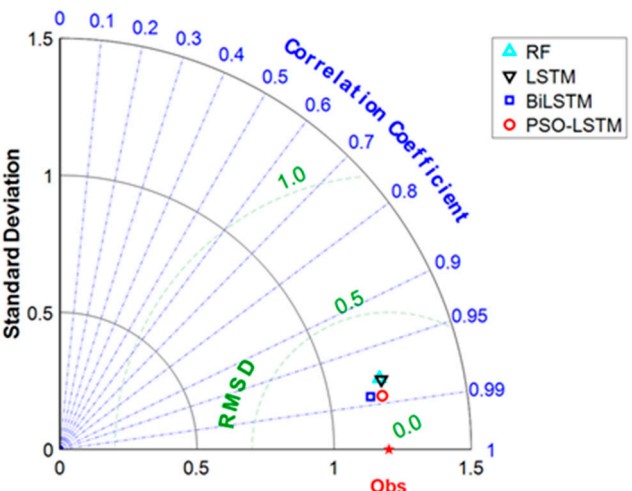

**Figure 9.** Comparison of the monthly average runoff simulation results of different models (Taylor diagram).

Glacial runoff is strongly influenced by the glacier, permafrost, the snow-related freeze–thaw process, and precipitation. Unlike general neural network models that can be used only as black-box models in the hydrological process simulation, the LSTM models introduce the cell state and gate structure. Owing to the cell state and gate structure, the LSTM models have certain hydrological characteristics, which allow them to learn information that should be saved or forgotten during the training process by transferring the time series data features to the next time steps when processing the state data and controlling the addition and removal of information through the gate structure. For instance, when an LSTM model is used to simulate glacier hydrology, the state characteristics of the time series data can be understood as the freeze–thaw state, basin storage, groundwater storage, soil water content, snow depth, and other hydrological state variables in the active layer. In addition, the gate structure can indicate the growth, outflow, and depletion of each state variable. When temperature and precipitation are used as input data, the LSTM model has a certain physical meaning while improving the model prediction accuracy.

In this study, the PSO algorithm was used to optimize important parameters of the LSTM model, such as the learning rate and the number of hidden-layer nodes, and the LSTM structure was improved by introducing the BiLSTM model, which represents a combination of forward and backward LSTM models. The results indicated that the optimization of the LSTM model's parameters and structure had a positive effect on the prediction accuracy of the benchmark model. However, the structure of the proposed LSTM model is complex, and there are also other parameters that affect the prediction effect of the model, such as the number of iterations, batch size, learning rate, and the number of hidden-layer nodes.

In this study, the benchmark model was improved through parameter and structural optimization, and certain progress in prediction effect was achieved. Still, it is necessary to study optimization algorithms of multiple model parameters further. In future work, hybrid multi-model network structures could be considered.

## 6. Conclusions

Glaciers have received great attention because of their sensitivity to climate and their significance for the storage and regulation of important freshwater resources in the form of solid reservoirs to recharge arid and semiarid oasis ecosystems. In this study, the correlation between climate factors and glacier-derived runoff was analyzed using multi-year data on the runoff, air temperature, and precipitation in glacier areas. Four models were constructed to predict the glacier-derived runoff.

In the past 26 years, air temperature showed an insignificant increasing trend, while the rainfall and glacially derived runoff showed insignificant decreasing trends.

In this study, four prediction models were constructed by inputting precipitation and air temperature two-dimensional data from two permafrost basins for the characteristics of alpine glacier areas. The results indicated that the PSO-LSTM and BiLSTM models outperformed the benchmark LSTM models in terms of MAE, RMSE, and $R^2$ evaluation indicators. Compared with the benchmark model, the root-mean-square error of the PSO-LSTM and BiLSTM models was reduced by 23.79% and 13.84%, respectively, and the mean absolute error was reduced by 13.74% and 4.25%, respectively. This proves the efficiency and correctness of the PSO algorithm and the BiLSTM model. However, there is a continuous requirement for the optimization of model parameters, model structures, and hybrid prediction models. The results of the glacier-derived runoff prediction models in this study will be of great significance for medium- and long-term hydrological prediction of high-coverage glacier watersheds in arid regions, as well as for urban development, agricultural irrigation, and reservoir allocation in downstream areas.

**Author Contributions:** This research article is the joint work of five authors. Conceptualization, B.M.; data curation, X.Y. and B.M.; formal analysis, X.Y. and B.M.; funding acquisition, B.M. and Z.S.; investigation, B.M. and M.S.; mythology, X.Y., B.M., and M.S.; project administration, B.M.; supervision, B.M.; validation, X.Y., B.M. and M.S.; writing—original draft, X.Y., L.N. and B.M.; writing—review and editing, B.M. and M.S. All authors have read and agreed to the published version of the manuscript.

**Funding:** The research presented in this article is financially supported by the National Natural Science Foundation of China (Grant No. 41762019 and grant No. U1603241).

**Institutional Review Board Statement:** Not applicable.

**Informed Consent Statement:** Not applicable.

**Data Availability Statement:** All data generated or analyzed during this study are included in this article.

**Acknowledgments:** The authors would like to thank who assisted long and strenuous hours to collect field data. We also thank the Key Laboratory of Oasis Ecology Xinjiang University for their support in providing data and discussion during the research. We are extremely thankful to an anonymous reviewer and editor for valuable comments that have greatly improved the clarity of this article.

**Conflicts of Interest:** The authors declare that there is no conflict of interest regarding the publication of this article.

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
