# Peer review of "Prediction of Glacially Derived Runoff in the Muzati River Watershed Based on the PSO-LSTM Model"

_water, doi:10.3390/w14132018_

Round 1
Reviewer 1 Report
The topic addressed is interesting for the journal. However, some sections of the current version of paper should be improved. The reviewer suggests a major revision. The article should be modified taking into account the suggestions indicated below.
- In Section 3, authors are encouraged to provide more details on BP, CNN and RF models with the parameters considered. In addition, it is not clear to me which parameters are considered for the LSTM model without PSO, e.g.: number of hidden layers, number of cells for each layer, optimization method, bath size, learning rate and number of training epochs. Highlighting the differences between the individual LSTM and PSO-LSTM would better highlight the novelty of the paper. For more details:
- RF: Razavi-Termeh, S. V., Abolghasem Sadeghi-Niaraki, and Soo-Mi Choi (2019). Groundwater Potential Mapping Using an Integrated Ensemble of Three Bivariate Statistical Models with Random Forest and Logistic Model Tree Models. Water, 11(8), 1596, doi: 10.3390/w11081596.
- LSTM: Granata, F. and Di Nunno, F. (2021). Forecasting evapotranspiration in different climates using ensembles of recurrent neural networks. Agricultural Water Management, 255, 107040, doi: 10.1016/j.agwat.2021.107040.
- CNN: Ghimire, S., Yaseen, Z.M., Farooque, A.A., Deo, R.C., Zhang, J. and Tao, X. (2021). Streamflow prediction using an integrated methodology based on convolutional neural network and long short-term memory networks. Scientific Reports, 11, 17497, doi: 10.1038/s41598-021-96751-4.
- Authors are encouraged to improve Figures 4 and 7. They seem of lower quality compared to Figures 5 and 6. It would also be advisable to standardize the style of the figures (text font, text size, etc ...). For example, in Figure 4, units may be reported in parenthesis: (m3/s), (°C), (mm).
- Authors are encouraged to discuss the exceptionality of the results obtained with the LSTM and PSO-LSTM models, with the latter that led to r-squared equal to 0.999 for the calibration/training stage. Moreover, another strange result can be observed for CNN, which showed better metrics for the validation stage with respect to the calibration stage.
- Based on the conclusion section, what is further research direction?
Author Response
Reviewer`s comments and suggestions for authors (reviewer-1):
The topic addressed is interesting for the journal. However, some sections of the current version of paper should be improved. The reviewer suggests a major revision. The article should be modified taking into account the suggestions indicated below.
Thank you very much for your encouraging and comments about our manuscript. We have addressed all the points you raised and also consequently revised the manuscript. The detailed amendments are mainly as follows:
- In Section 3, authors are encouraged to provide more details……
We Considering that there is a certain degree of similarity in the design structure of neural network models, such as neural network models use neural cells as the basic units, and by connecting these neural unit elements to form various forms of network structures, the optimization adjustments of the models all take into account the same hyperparameters such as learning rate, batch size, number of hidden layer units, etc. Considering that optimization of any one of the neural network models and ignoring the other Therefore, the CNN and BP neural network models are deleted. The LSTM model and PSO-LSTM model are retained, and the BiLSTM model is added. The BiLSTM model is a collection of forward and backward LSTM models, which is structurally optimized relative to the baseline LSTM model. The RF model, which has obvious differences in design structure and parameter properties, is also introduced because the model is integrated from multiple decision trees, and the output results are aggregated and synthesized by multiple evaluators, which are less affected by hyperparameters and have more stable output results. Therefore, the four groups of models in this modification are: RF, LSTM, PSO-LSTM, and BiLSTM models. The purpose of the adjustment is to highlight the PSO-LSTM model and the structure-optimized bidirectional BiLSTM model of the hybrid particle swarm algorithm parameter seeking on the benchmark LSTM, and the parameters of each model are introduced as follows:
We rewrote the details of each model and marked in yellow in the revised paper (Chapter 4.2)
- Authors are encouraged to improve Figures 4 and 7. They seem of lower quality compared to Figures 5 and 6. It would also be advisable to standardize the style of the figures (text font, text size, etc ...). For example, in Figure 4, units may be reported in parenthesis: (m3/s), (°C), (mm).
Thank you very much for your comments, We have adjusted the picture details in the revised paper (see in figure 4 and 7)
- Authors are encouraged to discuss the exceptionality of the results obtained with the LSTM and PSO-LSTM models, with the latter that led to r-squared equal to 0.999 for the calibration/training stage. Moreover, another strange result can be observed for CNN, which showed better metrics for the validation stage with respect to the calibration stage
Thank you very much for your comments, We rediscuss the variability in the results obtained for the benchmark LSTM, PSO-LSTM, and BiLSTM models and marked in yellow in the revised paper (Chapter 5.2)
In the question 1, the overfitting phenomena and solutions of CNN and PSO-LSTM models are discussed, and the modified results can be found in the fourth table of the revised paper.
- Based on the conclusion section, what is further research direction?
Thanks to the reviewer`s opinion, Subsequent studies were rewritten, and marked in yellow in the revised paper (Chapter 5.2)
Thank you very much for your time spent in reviewing our manuscript. We really appreciated for your help and your valuable comments that have greatly improved the clarity of this article.
Best regards.

Reviewer 2 Report
- Why is more than 80% of the time series used for training, whereas less than 20% is used for testing?
Please describe this in the literature as LSTM largely suffers from training data overfitting.
- Why BP, RF, and CNN are used for time series analysis and comparison with the LSTM-based model?
It is well understood in the data-sciences community that RNN-based models are optimal for time-series prediction and forecasting. It would be better if authors select some variations of LSTM like stacked, bidirectional, or CNN-LSTM and compare them with PSO-LSTM.
- Why linear and MK trend analysis are selected at the same time? What is the author's hypothesis on meteorological data w.r.t parametric and non-parametric?
- Please present training accuracy w.r.t training epoch.
Author Response
Reviewer`s comments and suggestions for authors (reviewer-2):
Thank you very much for your suggestion. We appreciate the time and effort you have spent to share your insightful comments, which will be seriously considered and adequately implemented. We will revise according to the comments. And the special features of PSO-LSTM and BiLSTM models based on the benchmark LSTM model are emphasized in the revised paper. The detailed revisions according to the revisions are mainly as follows:
- Why is more than 80% of the time series used for training, whereas less than 20% is used for testing? Please describe this in the literature as LSTM largely suffers from training data overfitting.
Thank you very much for your comments, We consider that whether it is machine learning, neural network learning, or process-oriented physical modeling, the setting of model parameters plays a crucial role in the model effect. Therefore, in the process of "localizing" the model, it is necessary to train the model with more data, so that the model parameters can be iteratively optimized over a larger sample size, and finally the model achieves the optimal global simulation effect over a larger data size. Therefore, in this paper, about 80% of the data are used for training the model, and about 20% of the data are used for validating the model.
In terms of overcoming model overfitting, we explain in the third paragraph of section 4.2 of the revised paper.
- Why BP, RF, and CNN are used for time series analysis and comparison with the LSTM-based model?
It is well understood in the data-sciences community that RNN-based models are optimal for time-series prediction and forecasting. It would be better if authors select some variations of LSTM like stacked, bidirectional, or CNN-LSTM and compare them with PSO-LSTM..
Thank you very much for your advice and suggestions. we consider that RNN has memory function and can discover the sequence relationship between samples, which is the preferred model to deal with time series, but when the input sequence length is too long, the role of data farther away from the current moment for the current moment is diminished, resulting in the network structure can not learn the sequence data of the more distant moment, in order to solve the RNN structure in dealing with long term memory when there is gradient vanishing ,this paper adopts the LSTM model of RNN improvement model, which introduces cell states and uses input gates, forgetting gates, and output gates to maintain and update cell states, which can reduce the possibility of gradient vanishing in RNN when processing long term memory, thus learning longer time series.
In the model selection and comparison, this modification takes into account that BP, CNN and LSTM all belong to the neural network class of models, which mostly use neural cells as the basic units of neural networks and form various forms of network structures by connecting these neurons, therefore, among these neural network models, only hyperparameter optimization or structural optimization is performed for a certain neural network model to compare and study other models with default parameter settings, the conclusions drawn are not universal. Therefore, this revision removes the BP and CNN neural network models and retains the LSTM model and the LSTM model with particle swarm optimized hyperparameters, and also adds the bidirectional LSTM model, RF. mainly to highlight the improvement in prediction accuracy of the PSO-LSTM model and BiLSTM with parameter optimization and structural optimization on the basis of the baseline LSTM model.
We rewrote the details of each model and marked in yellow in the revised paper (Chapter 4.2)
- Why linear and MK trend analysis are selected at the same time? What is the author's hypothesis on meteorological data w.r.t parametric and non-parametric?
Thank you very much for your comments and your advice, The main purpose of introducing linear and MK trend analysis of meteorological data is to discuss as a separate analysis the strange phenomena of the past 26 years, such as the increase in temperature and rainfall that did not lead to an increase in flow, and to introduce linear and MK trend analysis not only to derive trends in temperature and rainfall (e.g., temperature increases by 0.42 °C and rainfall increases by 49.6 mm per decade), but also to verify whether the conclusions reached are statistically significant. 49.6 mm), but also to verify whether the conclusions reached are statistically significant, e.g., a trend of significant increase in temperature at the 0.05 level and a trend of non-significant increase in rainfall at the 0.05 level in combination with the MK trend test.
The MK (Mann-Kendall) trend test does not target specific parameters and does not make strict assumptions about the distribution of the variables, so it is also known as an arbitrary distribution test or a nonparametric test, and is therefore commonly used to test the trend of a time series.
- Please present training accuracy w.r.t training epoch.
Thank you very much for your comments. we illustrate the loss function values and RMSE for the training process. in the revised paper (In section 4.3 of the revised paper)
Thank you very much for your suggestion. We appreciate the time and effort you have spent to share your insightful comments, which will be seriously considered and adequately implemented. We have addressed all the points you raised; it's been very helpful to improve the English and content of the manuscript. We have addressed all the points you raised, and please see the revised paper for details (which are highlighted).
Best regards.

Round 2
Reviewer 1 Report
The reviewer congratulates the authors on improving the paper taking into account all the requests of the reviewers
Author Response
Editor`s comments and suggestions for authors
Dear authors, first of all, i would like to thanks for considering the comments of the reviewers. still i have some additional comments, which need to be addressed before i can recommend the manuscript for publication. i agree with the reviewers that the topic is interesting, relevant and within the scope of the journal.
my main general comment is related to the justification and support of the interpretations made from the presented results. the manuscript shows that in the study region (station at the outlet) there is no statistically significant change in precipitation and runoff, but the air temperature has increased. the explanation/attribution given in the manuscript is that the main controlling factor is the shrinkage of mountain glaciers. this is however not documented by some observations. an alternative explanation could be that even the annual air temperature increases, the absolute value of increased air temperature is still below zero, so have no effect on glacier melt. some additional analyses here are needed to support the interpretations made. what are the trends of monthly air temperatures, precipitation and runoff?
the second aim of the manuscript is to propose/develop/evaluate a neural network model for glacier melt prediction. here i missed a more comprehensive review of the international literature about this research subject. which ann and artificial intelligence-based methods have been already applied in the past for glacier melt predictions ?what are still the research gaps and how this study contributes to some novel knowledge? this needs to be clearly formulated. it will be interesting to see some added value of using neural networks for melt prediction. the monthly runoff regime seems to be quite stable over the years. what is the advantage of using neural networks e.g. compared to using only long-term mean monthly values for predictions?the topic is clealry hydrological, so please add some more hydro-logical interpretations of the results, i.e. not just providing some technical explanations about the methods and efficiency, but also some attribution of the benefits form the point of view of hydrological processes and mechanisms of glacier melt generation. finally, please do a very careful proof of language. there are some uncomplete sentences or parts which are difficult to understand. i believe that addressing these comments will improve the overall quality of the manuscript and presented results and itnerpretations.
Response to the editor: The authors would like to thank the editor for reviewing this work and put providing valuable comments and suggestions. The manuscript has been revised according to the comments. The detailed amendments are mainly as follows.
- Analysis and interpretation of temperature, precipitation, and runoff.
The authors would like to thank the editors for pointing out the shortcomings in the analysis presented in the manuscript. For instance, the high-coverage glacier basin studied in this study ranges from (80°20′0″–81°0′0″E, 41°50′0″–42°20′0″N); if the previous CRU weather dataset is selected, there is a problem that the spatial resolution of the original CRU released weather dataset is coarse, approximately 0.5° × 0.5°. The effective meteorological grid points covering the study area are about 1–2, and the effective data are too scarce, which makes the extracted temperature and precipitation in the study area have certain errors. The ERA5-Land meteorological dataset with a higher accuracy is used in this revision, which has a spatial resolution of approximately 11 km2 × 11 km2 and contains approximately 22 effective meteorological grid points in the study area so that the spatial differences in the study area can be considered.
In the revised manuscript, the meteorological dataset with a high spatial resolution has been introduced (lines 135–141 in red section); the monthly average temperature and precipitation have been analyzed (lines 282287 in red section); the discussion section has been revised (lines 390–393, 422–433 in red section).
- which ANN and artificial intelligence-based methods have been already applied in the past for glacier melt predictions? what are still the research gaps and how this study contributes to some novel knowledge? this needs to be clearly formulated
Thank you for your comment. The related literature has been reviewed, including the application of artificial neural networks to glacier runoff simulations (lines 76–87 marked in red). The characteristics of the current application of artificial neural networks in this area have been analyzed and discussed, as well as the contributions of this study in improving the existing neural network models (lines 449–453 marked in red).
- what is the advantage of using neural networks e g. compared to using only long-term mean monthly values for predictions? the topic is clearly hydrological, so please add some more hydro-logical interpretations of the results i.e. not just providing some technical explanations about the methods and efficiency, but also some attribution of the benefits form the point of view of hydrological processes and mechanisms of glacier melt generation.
Thank you for your comment. In the revised manuscript, the advantages of the neural network model have been highlighted (lines 454–462 marked in red). In addition, more hydrological implications of the model have been presented (lines 478–492 marked in red).
